# Forest Recreational Services in the Face of COVID-19 Pandemic Stress

**Dastan Bamwesigye** [1], **Jitka Fialová** [2,*], **Petr Kupec** [2], **Jan Łukaszkiewicz** [3]
**and Beata Fortuna-Antoszkiewicz** [3]

1 Department of Forest and Wood Products Economics and Policy, Faculty of Forestry and Wood Technology, Mendel University in Brno, Zemedelska 3, 613 00 Brno, Czech Republic; xbamwesi@mendelu.cz
2 Department of Landscape Management, Faculty of Forestry and Wood Technology, Mendel University in Brno, 613 00 Brno, Czech Republic; petr.kupec@mendelu.cz
3 Department of Landscape Architecture, Institute of Environmental Engineering, Warsaw University of Life Sciences—SGGW, ul. Nowoursynowska 159, 02-776 Warsaw, Poland; jan_lukaszkiewicz@sggw.edu.pl (J.Ł.); beata_fortuna_antoszkiewicz@sggw.edu.pl (B.F.-A.)
* Correspondence: jitka.fialova@mendelu.cz

**Abstract:** Forest ecosystems provide numerous services and benefits to both humans and biodiversity. Similarly, urban forests services play a vital role by providing urban dwellers with recreational and leisure space, mental health relief, and meditation. In the wake of the COVID-19 epidemic, many people living in the urban areas could benefit from the forest and park recreational services to relieve psychological stress due to lockdown rules. The study examined existing literature simultaneously; however, very few studies have presented the relationships between forest services' role on COVID-19 stress relief. Furthermore, we examined forest visitors' frequency at the Training Forest Enterprise (TFE) Masaryk Forest Křtiny in the outskirts of Brno City in the Czech Republic. The study collected data using a TRAFx infrared trail counter before the pandemic (2015–2018) and during the COVID-19 period (2021). As in other studies of the subject, we observed an increasing trend in forest visits during the COVID-19 lockdown in 2021, compared to the same months before the pandemic in 2016 and 2017. We recommend further research to focus on scientific analysis of the relationship between forest ecosystem services and COVID-19 stress and mental health. Moreover, given the spike in visitors during the COVID-19 lockdown in 2021 in March and April, our data provide evidence regarding the role of nature for relieving stress and supporting mental and physical health. Policy, decision-makers and medical advisors could use such data and study to guide future lockdowns and pandemic situations regarding nature and forest recreational use and importance.

**Keywords:** urban forest services; recreation; social health; stress resilience; COVID-19 pandemics; vitamin D

## 1. Introduction

The COVID-19 pandemic had claimed over 1 million dead and over 30 million infections globally by the end of 2020 [1–6]. Mortalities continued to soar in the year 2021, with over 100 nations experiencing severe death [7–12]. As of 1 December 2021, more than 5.2 million people have died from the same pandemic [12].

The increase in death and infections were also followed by many other socioeconomic and environmental problems including loss of jobs, strict lockdowns, and competition on some of the resources including food, water, and energy sources. These occurrences have left many people worldwide with mental health and psychological welfare issues [13–20]. The constant lockdowns that are practically necessary have also compounded the already bad situation and increased mental health rates across the globe [13–16]. Therefore, herein we assessed the role of nature and forest recreation services during the pandemic. We studied data before the pandemic and during the pandemic to understand the trend and to support our hypothesis of the role of nature on mental health [21].

The healthy and lush greenery has long been known as a factor that exerts a relaxing, soothing, and therapeutic effects on the human body (both physically and mentally). Throughout the 19th century, greenery was consciously incorporated into cities' urban structures (e.g., urban forests, forest parks, woodlots etc.), intensifying such activities in the twentieth and twenty-first centuries [22,23].

The positive role of greenery, especially urban forests, on city dwellers' social health and mental condition cannot be overestimated [3,4,6,24,25]. Around 2012, the WHO had predicted that untreated mental health issues would be the leading cause of morbidity and mortality worldwide by 2030 [25]. Unfortunately, it looks like the COVID-19 pandemic may significantly accelerate this phenomenon, with tremendous emotional costs on a global scale and a negative impact on the quality of life for most of the human population [6]. These include, for example, a widespread, overwhelming sense of loneliness and mental disorders, which lead to severe and deadly diseases like depression. With the prolonged stress caused by "lock-downs", not only the human psyche, but also immunity to diseases and the body's overall activity on a biological level may be affected—a fact long known before the current pandemic [2,13,14].

In this context, recent studies confirm that vegetation and urban green spaces (e.g., parks, municipal forests, woodlots, etc.) have a remarkably beneficial effect on human health (soma and psyche). Self-esteem, life satisfaction, and subjective feelings of happiness are significantly related to the frequency of green spaces, even to such a seemingly trivial factor as whether such spaces can be seen from apartment windows. Vegetation can significantly mitigate risk factors for human mental and physical health [13,14,26–28]. It is especially crucial for people living in cities and urban areas already inhabited by most of the global population (projected to be 5.0 billion in 2030) [28–31].

Urban forests located closely around the cities are traditionally positively influenced factors to mitigate climate changes, negative disturbances in water management, impoverishment of the animal and plant world, etc. However, to fulfill their function correctly, such green areas must be adequately shaped, maintained, and protected in a multi-year process [22,23]. Therefore, this publication hypothesizes that urban forests are vital for urban environments, urban society and the quality of peoples' life and health, especially during the pandemic. For this reason, there is a rising global importance to shape greenery, such as urban forests, to obtain its maximum pro-health impact.

The identified problems of the subject in this paper helped to understand how the intensity and frequency of forest visitors and recreational traffic develop in the selected urban forest over the following years and the period of the COVID-19 pandemic. Hence, this provides a basis for decision making for improvements for forests and other urban recreational areas and lockdown-informed decision making considering the role of nature in the period.

## 2. The Impact of Urban Forest on Urban Environment and Quality of Life

Issues identified during the first stage of data collection, i.e., the literature review, aimed to describe how urban green areas, especially urban forests, can play a particularly important and vital role in the quality of recreation and social health of people living in cities, especially during a pandemic. It must be stated from the outset that there has long been broad evidence of the powerful effects of plants on human quality of life, e.g., in the aesthetic, emotional or physiological spheres.

It has long been known that woody plants (e.g., those in forests) occurring around urban areas are the simplest (and oldest known) way to reduce air pollution. In such a context, the phytoremediation is the predisposition of plants to reduce pollution generated by industry, home heating devices, vehicle traffic, e.g., by filtration and reduction of the concentration of particulate matter (PM) and gases (e.g., nitrogen oxides $NO_x$, carbon monoxide $CO$, sulfur oxide $SO_2$ and ozone $O_3$) [32–34].

The European Environment Agency (EEA) indicates that critical amounts of particulate matter (PM) in the atmosphere are a global problem (cause of disease and death). Apart

from rainfall, the only effective form of reducing air pollution by PM are plants—and especially trees—that can directly accumulate these pollutants on the surfaces of leaves (needles and blades), young shoots, and wax-saturated bark. Trees can indirectly reduce dust pollution through transpiration and influence, changing climatic conditions [33–35]. The mitigating effect of trees is significant for ecology and visibility, especially compared to the effectiveness of various technical solutions, such as energy-saving technologies, reduction of gas emissions and air dust, technical elements of blue and green infrastructure, and others [33,34].

Properly composed and maintained greenery (urban forests among others) located around areas with residential functions is essential to ensure appropriate environmental conditions and sustainable revitalization of the urban structure [22]. In general, the natural impact of green systems (parks, urban forests, etc.) includes stimulation of air mass exchange and purification, reduction of the greenhouse effect through carbon dioxide assimilation and oxygen release, reduction of air temperature amplitudes (reduction of "heat islands"), improvement of soil structure, water retention, noise suppression, etc. [23,33,34].

On the other hand, the scope of the positive influence of greenery on the human body is well documented and genuinely impressive. This notion manifests itself by, e.g., lowering the stress level of people at leisure, improving social interactions in public spaces, speeding up the recovery process of the ill (e.g., from greenery around hospitals, sanatoria, and health centers), reducing mental fatigue, improving concentration and performance (e.g., from greenery in the area around schools, preschools, workplaces), and suppressing feelings of aggression and violence [23,33,35–39].

Research on the health-promoting effects of vegetation on the human body has been put to practical use for years, e.g., in Japan as part of therapies known as "forest bathing" (jap. "*shinrin-yoku*"). For this reason, attempts to improve the quality of life in cities by using the health-promoting effects of greenery and forests on whole social groups and individuals are multiplying around the world [22,29,30,33,39,40].

The civilizational changes in city dwellers' lifestyle translate into a strongly felt social need for recreation and leisure-specific for each epoch and time [41]. During a COVID-19 pandemic, it is of great importance to provide high-quality recreational places (for everyday and holiday recreation), most often close to places of residence due to the intense pace of life and increasing communication difficulties [23,39,42,43].

In the context of the pandemic, the recreational importance of urban parks and forests is growing immeasurably. In this case, exceptionally tall green (trees) can significantly stimulate the feeling of the so-called comfort or, in other words, "well-being" and exert an influence on the so-called "recreational bioclimate" of biological factors determining the quality of recreation. Depending on their area and ecological diversity, planting trees can significantly modify the bioclimatic conditions locally and in neighboring areas. In such a case, the greenery structure has the decisive significance (spatial structure, species structure, age structure) obtained through long-term shaping and care; it is visually attractive and ensures optimal light and thermal conditions, ventilation, atmospheric air composition, etc. [23,39].

It is worth mentioning that, during the COVID-19 pandemic, the therapeutic impact of urban forests on social health became significant due to the favorable availability of solar radiation. Among people vacationing in the forest, the natural possibility of skin synthesis of vitamin $D_3$ (the so-called "sun vitamin") increases, which is why it has long been recognized that the availability of sunlight, such as during recreation in a luminous forest with a loose canopy, is of great health-promoting importance, including in relation to the COVID-19 pandemic [23,44–47]. The importance of an adequate vitamin D supply cannot be overestimated, especially during the COVID-19 pandemic. Presently, the "sun vitamin" is regarded as an essential, though not a sufficient, factor for the proper functioning of the human body's key physiological pathways of cells [48–59]. Existing data suggest that patients showing a mild course of infection also show sufficient vitamin D supplementation

levels, while the group with the acute and critical course of infection is generally poorly supplied with vitamin D [60–62].

## 3. Materials and Methods

The identification of the issues and the formulation of the main goals of the research allowed us to start the first stage of work. The substantive scope of the study was determined by resolving problems contained in the informative questions (addressed to the literature on the subject), which were formulated as follows:

- How can the forest environment affect human physical and mental health (especially important during the COVID-19 pandemic)?
- What factors determine favorable recreational conditions in the forest?

Next, the obtained answers to the above questions constitute the basis on which the authors formulated the following research questions (concerning the data collected during field research and observation):

- What are the dynamics of visiting an urban forest?
- Is there a relationship between attendance and external factors (season, holidays, COVID-19 pandemic)?

The first stage of the research contained an extensive literature search that was conducted to compile examples of the importance of urban forests for the quality of recreation of urban communities. The data collected at this research stage was from the literature and the authors' own professional, scientific, and practical experience.

The second stage of the research was field data analyses collected in subsequent years 2015–2018 and 2021 in Training Forest Enterprise (TFE) Masaryk Forest Křtiny. Forest visitors' data results before and during the COVID-19 lockdown crisis were recorded at a selected point at the entry to the forest when available [21].

## 4. Field Research

**The research area** was a Training Forest Enterprise (TFE) Masaryk Forest Křtiny [63]. It is an organizational part of Mendel University in Brno (Czechia) and a special-purpose facility of its Faculty of Forestry and Wood Technology. Forestland property has an area of 10,265 ha. Forests form a continuous complex that immediately links with the northern limits of the Moravian metropolis of Brno (Figure 1) and reaches as far as the town of Blansko. The forest is situated at altitudes ranging from 210 to 575 m above sea level. The forest is characterized by various natural conditions, which predetermined it as the place to establish the special-purpose facility of the university. The mean annual temperature is 7.5 °C, and the mean annual precipitation is 610 mm. The topography is broken with deep-incised valleys and glens, especially those of the Svitava River and the Křtinský Brook. Granodiorites, Culmian grawacks, and limestone form a parent rock. About a third of the TFE area is located within the protected landscape area of the Moravian Karst.

The main native forest-forming species of trees are spruce, pine, and larch for conifers, and beech and oak for broadleaves. Showing signs of regeneration, fir is gradually returning to the forest stands. The shelterwood system is successfully introduced for acidic sites of beech and oak and even in spruce stands. An effective attempt was also made to rebuild the stand towards a selective forest. As a result, TFE enjoys an entirely exceptional position in the fulfillment of the aesthetic and educational functions of the forest. There are whole parts of stands with the natural species composition left without intervention in the past. The natural beauties of local forests are intentionally maintained and improved.

Various places and objects which have come to existence in TFE Masaryk Forest Křtiny, as a result, make it more attractive for outings of hikers from nearby towns and villages. These include, for example, carefully maintained viewpoints located on elevations or forest glades established in the complex of continuous forests. The surrounding meadows are very colourful and visually attractive, due to the introduction of exotic tree species in the area. Forest springs are sought and looked after, and new fountains are built. A

particularly characteristic tourist attraction is the Forest Pantheon, which is a real-world rarity. It consists of a set of more than seventy monuments, fountains, and memorial tablets located in specially selected places devoted to distinguished foresters and artists who have dedicated their works to nature; in addition, there are games, trees and the forests themselves.

The main goal of monitoring forest visitors, tourism, and traffic is to provide basic information about the number of visitors and data on the temporal variability of traffic (within a day, week, months, year, and the seasons) and distribution of visitors within the target area. Monitoring forest visits in the recent period over protected areas in tourism helps plan administration activities and sustainable management.

We installed an automatic counter of hikers and bikers to monitor the selected representative forest hauling road in TFE Masaryk Forest Křtiny (Figures 1–3). The monitoring devices were installed in July 2014, and the monitoring was (with some interruption) conducted until July 2021. The trail visitor monitoring used automatic reader Pyro Box Compact from Eco-counter in years 2014–2018. A barrier was installed on the road where the counter was mounted, limiting the number of vehicles entering the trail. Cars were restricted to foresters and hunters.

This device counts all forest road users on the trail (hikers, bikers, in-liners, etc.) without distinguishing among them. Counting forest visitors is based on the temperature difference between a human body and its surroundings, including animals such as horses and dogs [64]. The readers can distinguish the direction of the movement and are installed in the narrowest places of the trails to prevent counting two persons walking side by side as one [64].

In the year 2021, a TRAFx infrared trail counter was installed. According to [64], it counts people on trails, paths, and sidewalks. It has an advanced microelectronic design and high-quality infrared scope. The counter's significant advantage is its large storage capacity (millions of counts), and long battery life. The TRAFx infrared trail counter counts people walking along this particular forest road, i.e., walkers, hikers, joggers, inline skaters, horseback riders, cyclists, etc. It senses and detects the infrared wavelength that people emit. Unlike other trail counters, it does not require a receiving unit or reflector to operate. It results in a very compact, unobtrusive design that reduces the risk of vandalism. The TRAFx infrared trail counter also works well in winter conditions along trails dedicated for snowshoe, ski, or snowmobile.

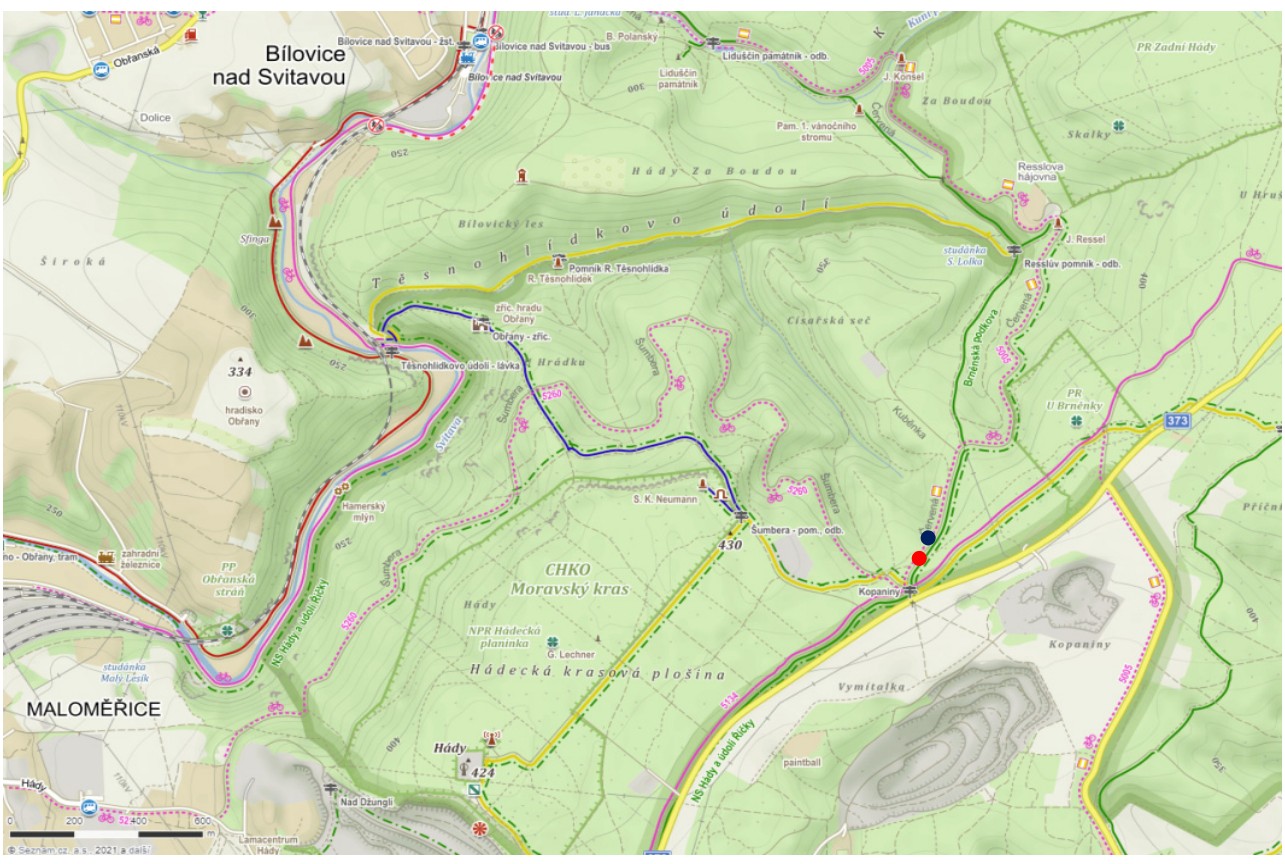

**Figure 1.** Evaluated part of Training Forest Enterprise (TFE) Masaryk Forest Křtiny (Red and blue dots on the forest road "Červená" show the places of counters, forests are in green color) [65].

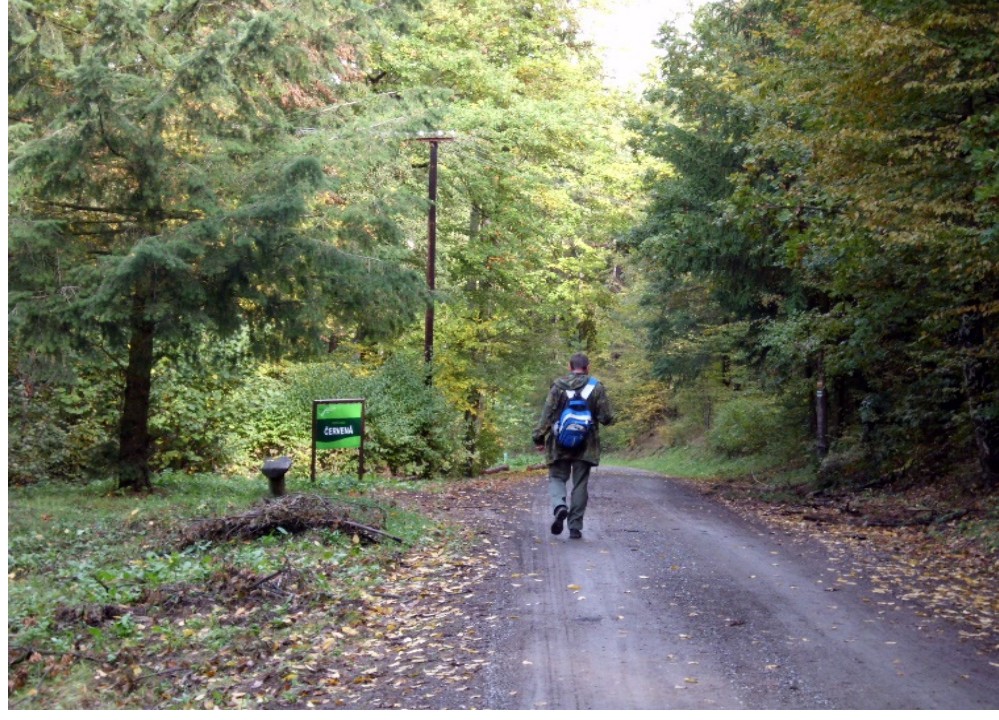

**Figure 2.** Training Forest Enterprise (TFE) Masaryk Forest Křtiny (forest road "Červená") (photo by: Jitka Fialová, 15 September 2017).

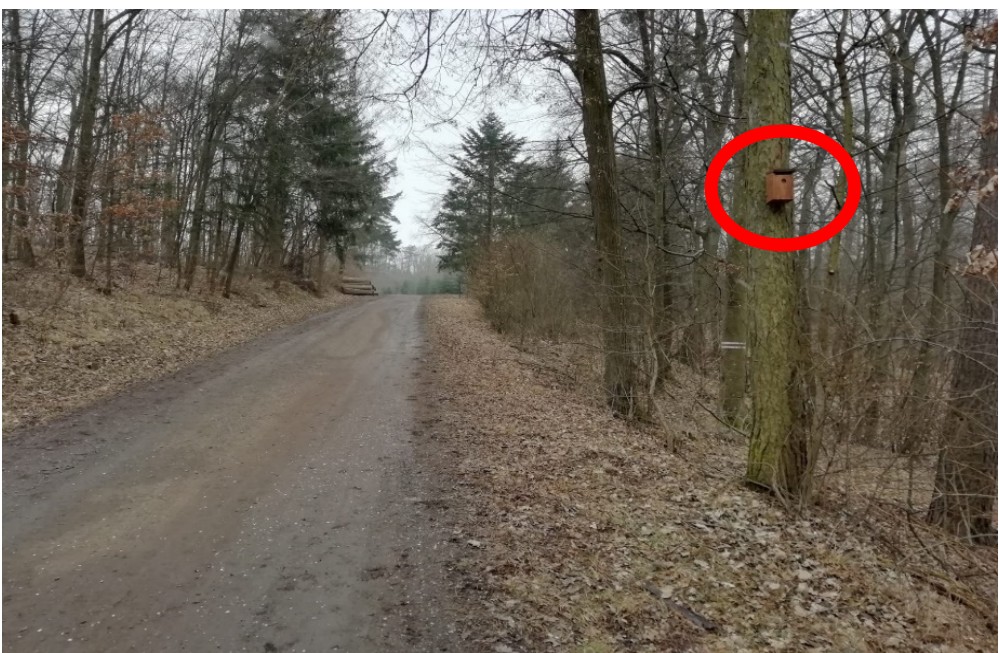

**Figure 3.** The TRAFx infrared trail counter counts people—walkers, hikers, joggers, inline skaters, horseback riders, and cyclists. Installed counter in the form of the wooden nest (in red circle) in the year 2021 (photo by: Jitka Fialová, 15 March 2021).

## 5. Results of Field Research

We present data collected over the years using an installed counter in the form of a metal box (July 2014–May 2018) and the form of a wooden nest (2021) on the forest road "Červená" (Red Trail) in the Training Forest Enterprise Masaryk Forest Křtiny near Brno City. The data collected is from July 2014 to 11 June 2018, when the counter was removed. Given the COVID-19 situation, we put the counter back to the same spot from the 1 March 2021 during Lockdown (Figures 1–3).

Our results showed a spike during March, April and May compared to June and July in 2021 (Figure 4). The strict and heavy COVID-19 lockdown had run from November 2020 to the end of May 2021. Compared to other years, the results showed a frequency spike during March, April, and May again for years 2015, 2016, 2017, 2018, and 2021 (Figures 4–6, Appendix A Figures A1–A5).

The COVID-19 lockdown had run from November 2020 to the end of May 2021. The data of March, April, and May 2021, compared to the same period of other years, on average shows a higher trend, apart from a few days in April and May, on which values of 1088 and 1312 forest visitors, respectively, were recorded on given days. During the lockdown, we observed significant values of 624, 595, and 873 on given days in March and 553,640, 427, 559 in April and towards the end of May (Figure 6).

During the warm period of the summer, the collected data show a declining trend of numbers of visitors in the forest, in each category. Due to the periods of good weather during the summer, our data illustrate more visits in the forest, but on a more stable level compared to very high spikes of the visiting frequency in the spring period. In addition, it is very interesting that there were more forest visitors in the summers of 2016 and 2017 than in 2021 (Figure 5, Appendix A Figures A1–A5).

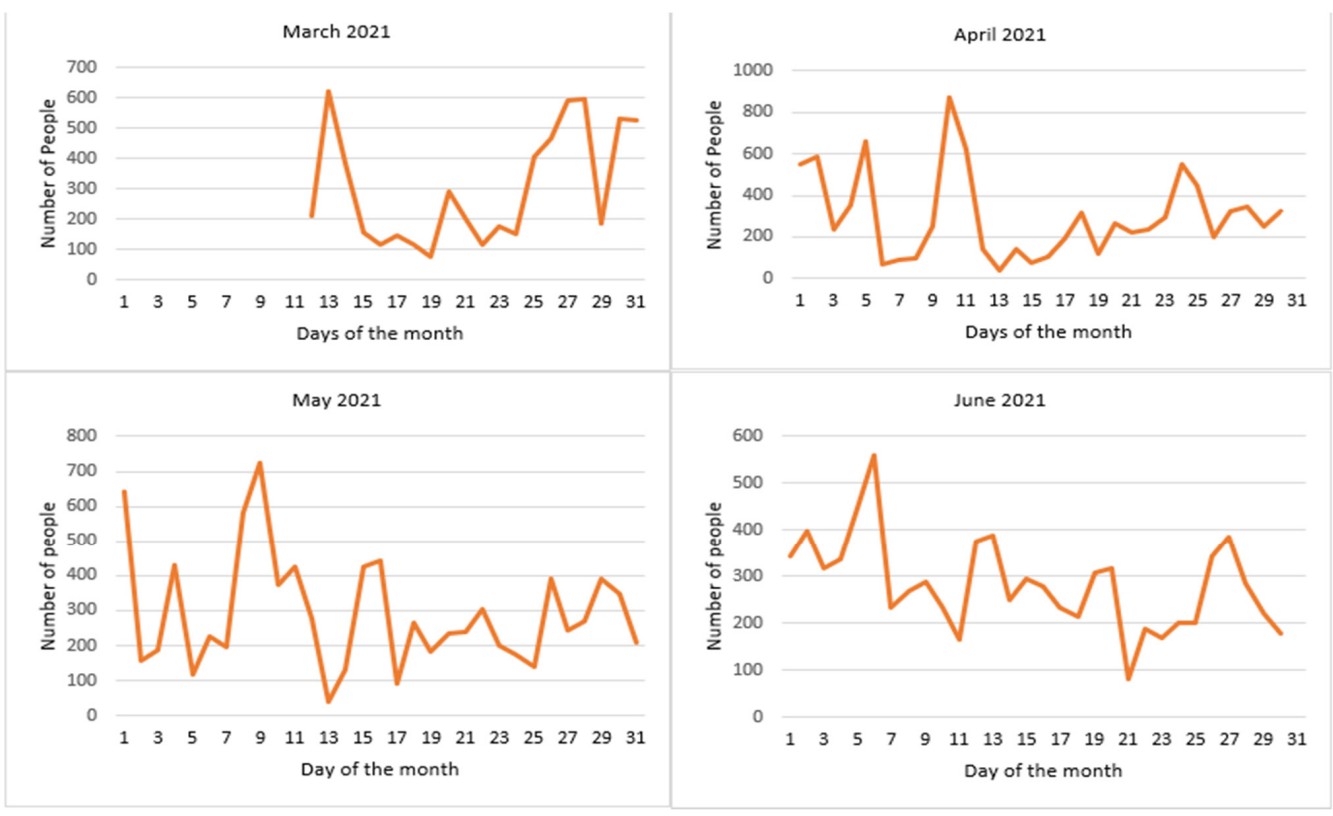

**Figure 4.** Recorded number of visitors per day in given months together in 2021.

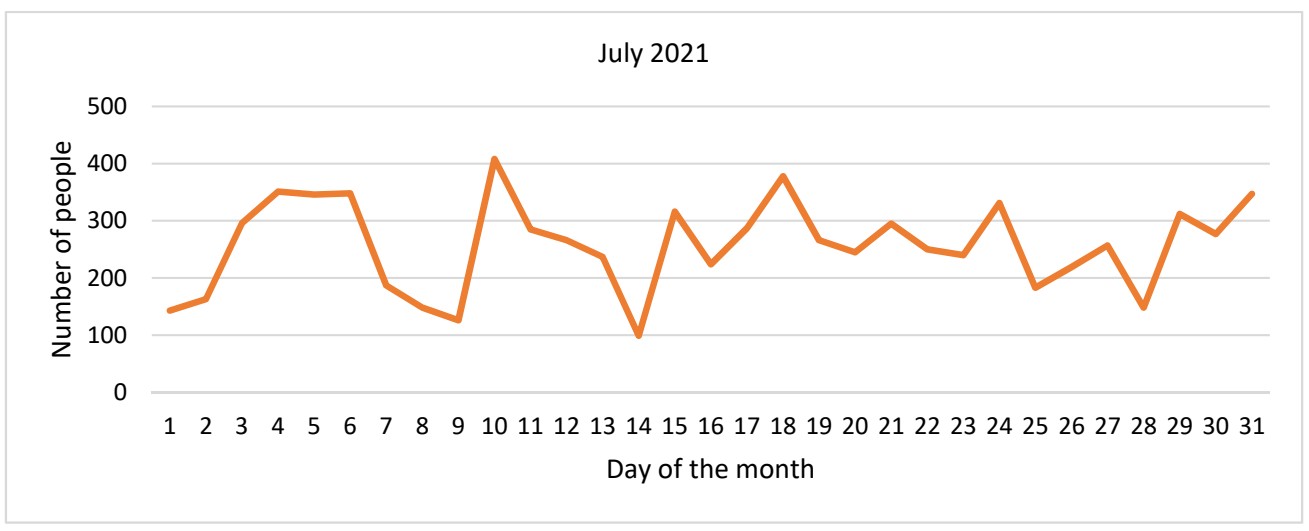

**Figure 5.** Recorded number of visitors per day in July in 2021.

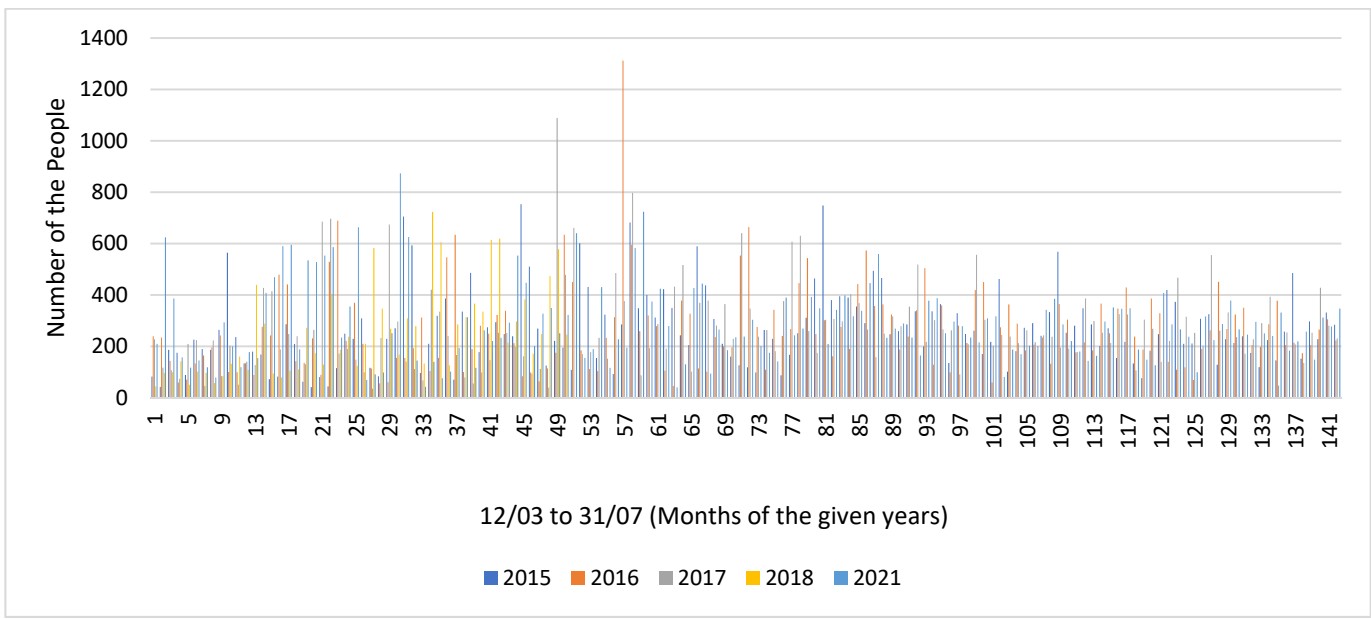

**Figure 6.** The frequency of visitors of Training Forest Enterprise Masaryk Forest Křtiny (Forest Road "Červená") in the subsequent years before and after COVID-19 (March to July).

In general, the studied months and years showed that, overall, 2021 had more total forest visitors at 40,616 people, 2017 at 38,502, 2015 at 37,820, 2016 at 37,491 and 2018 at 12,696 people, respectively (Figure 7). However, 2018 cannot be substantively counted regarding the total since June and July data are missing. The lower number of forest visitors in 2016 could be associated with other factors, such as socioeconomic characteristics and or climate. These factors could be studied in-depth to get to the gist of the discussion [66–72].

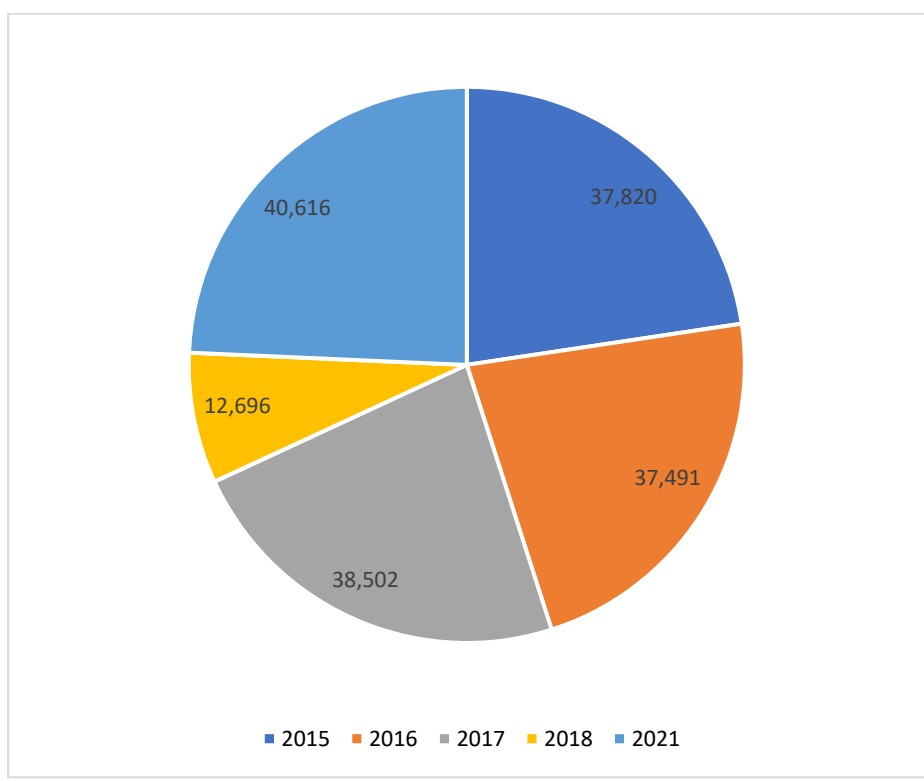

**Figure 7.** Total number of visitors in given years before and after COVID-19 (March to July).

We conclude that, ceteris paribus, the studied data showed a notable trend in the COVID-19 lockdown period with special attention in March, April, and May 2021. Even though previous years also recorded high visitors in the forest, the COVID-19 period showed exceptional results (Figure 6). Moreover, the year with COVID-19 presented the highest observed visitors, with more than 800 visitors than the second highest year. This trend is also reflected in the summary statistics of the selected data from March to July through the subsequent years (Table 1 and Figure 8). The mean and median scores showed high scores for 2021, 286 and 250 people, respectively. Understanding the ranges in this data can further be observed in the quartile ranges, which illustrated an almost equal distribution (Figure 8).

**Table 1.** Summary statistics of the frequency of visitors of TFE Masaryk Forest Křtiny (Forest Road "Červená") in the subsequent years.

| Year | Mean | Median | S.D. | Minimum | Maximum |
|------|------|--------|------|---------|---------|
| 2015 | 266.3 | 247.0 | 146.0 | 41.00 | 753.0 |
| 2016 | 264.0 | 235.0 | 166.5 | 45.00 | 1312.0 |
| 2017 | 271.1 | 239.5 | 164.2 | 35.00 | 1088.0 |
| 2018 | 253.9 | 212.0 | 173.7 | 44.00 | 723.0 |
| 2021 | 286.0 | 259.5 | 150.6 | 40.00 | 873.0 |

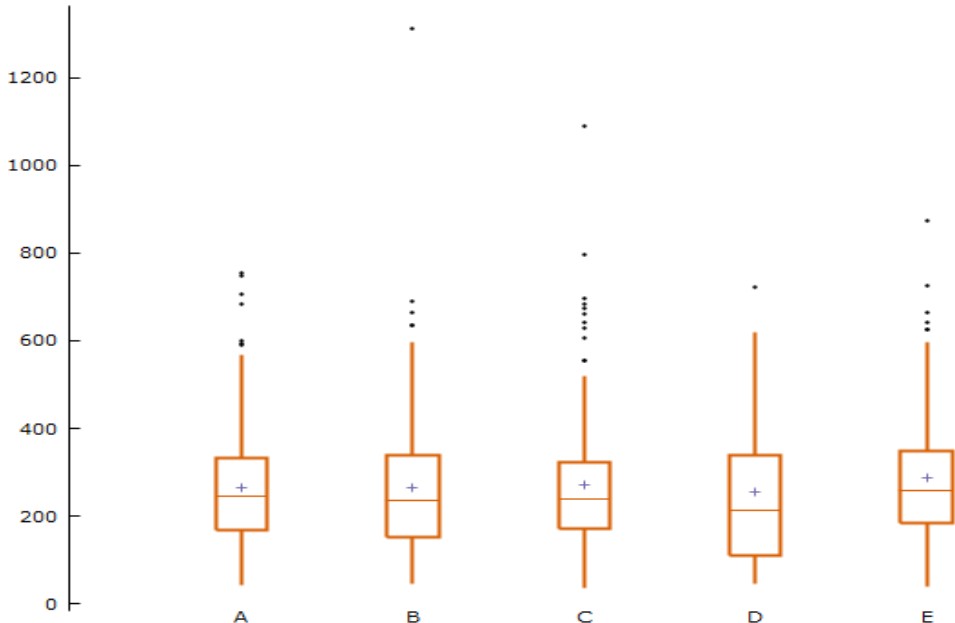

**Figure 8.** The box plot of mean visiting annual frequency (no. of people referring to Table 1) in TFE Masaryk Forest Křtiny (forest road "Červená") in the subsequent years before and after the COVID-19 pandemic (where A is 2015, B is 2016, C is 2017, D is 2018 and E is 2021).

## 6. Discussion and Conclusions

The trend of visiting frequency in the TFE Masaryk Forest Křtiny before and after the COVID-19 lockdown illustrates the positive meaning of forest recreation services' referring to the number of visitors in most periods (Figures 4–6). A similar trend is observed in analyzing the data of forest visits during the weekends in the same period studied. It confirms a greater spike during the lockdown period due to COVID-19 in March, April, and some parts of May (Figure 9). Even though the notable trend was observed more for weekends throughout the years (Figure 9), during the COVID-19 pandemic, relatively more visitors went to the studied forest of TFE Křtiny.

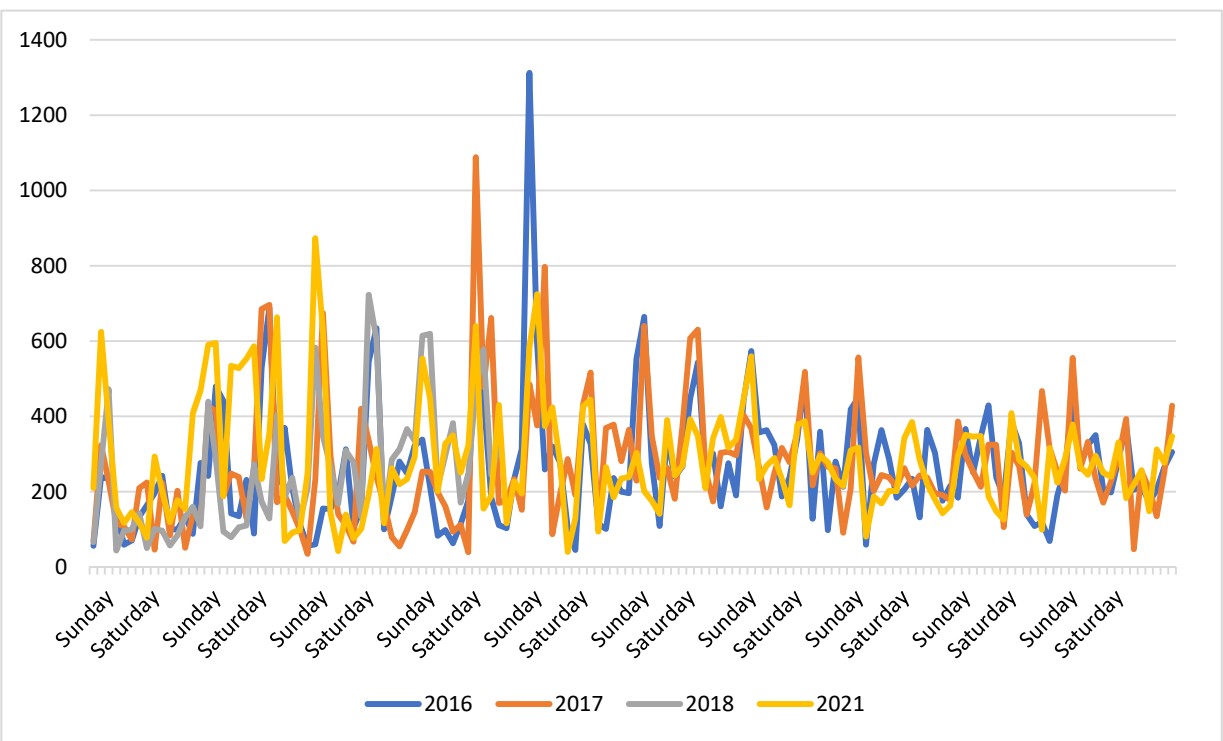

**Figure 9.** Forest visitors on the weekends from 2016 to 2018, and 2021 in months of March to July. (X-axis is months March to July in selected years, Y-axis is the number of forest visitors at of Training Forest Enterprise Masaryk Forest Křtiny).

The summary statistics did not show any significant disparity given a lower standard deviation than the mean and median (Table 1). Our results, in general, show higher visitor numbers in the forest on most days during the COVID-19 lockdown than in comparable periods from 2015 to 2018. It can also be confirmed by the mean and median number of visitors (Table 1) in subsequent periods (Figures 4–6 and Figure 8).

The number of visitors could be an illustrative measure of the recreational use of a forest. Furthermore, in economic terms, it is possible to estimate the scale of annual revenues related to the recreational use of the forest. Therefore, forest ecosystem services provide society with various non-monetary, but still essential, benefits, such as phytoremediation, recreation, positive effects on physical and mental health, the natural environment's beauty and aesthetics, and many others. In this case, exceptionally tall green (trees) can significantly stimulate the feeling of the so-called comfort or, in other words, "wellbeing" and exert an influence on the "recreational bioclimate" of the forest of biological variables determining the quality of recreation. Depending on their area and ecological diversity, forests can significantly modify the bioclimatic conditions locally and in neighboring areas. The decisive significance, in this case, is the structure of woodlots and stands (spatial, species, age structure)-visually attractive, ensuring optimal light and thermal conditions or ventilation etc. What is undoubtedly crucial during COVID-19 pandemics, forest recreation activities generally enhance peoples' wellbeing, especially those that enjoy a wide range of activities. Nonetheless, it is speculated that many people still lack enough knowledge regarding the benefits and costs associated with forest recreation. As a result, this usually affects their willingness to pay for forests' existence to ensure forest use-value sustainability, e.g., for recreation [66–71].

Moreover, while there is still limited information that gives a clear picture of recreation as one of the utilitarian values of the forest, there is no doubt that most of the studies analyzed in this chapter provide a detailed insight into the services and benefits of forest recreation. This information can assist forest management in strategic planning in integrating every vital aspect of forest service maintenance and benefits even in pandemic times [3,4,56,67–69].

However, limited research has been done to emphasize the role of forest ecosystems during this COVID-19 period. As much as numerous studies have been conducted on COVID-19 and its relation to mental health and stress, very few studies have linked COVID-19 epidemic stress to forest ecosystem services, given the latter's role. Of course, there were concerns over the risk of spreading the disease from one person to another. Nevertheless, [2,3,26,27] showed a sharp decline in the pandemic, while forest visitors showed a sharp increase.

As Weinbrenner et al. [72] mentioned, many participants describe that they visited the forest to cope with psychological stress caused by the COVID-19 pandemic, such as fear of infection, de-limited working hours due to home office solutions, and a lack of social contacts. For many forest visitors, the forest has become a place of compensation. Many young participants no longer met with friends in cafés, but in the forest. Gyms were moved into the forest. Off the trails, people used tree stumps to meditate on them. By providing many opportunities for very different activities, the forest serves as a substitute- or functional equivalent-on many levels. However, new fields of conflict emerged which are specific to the COVID-19 pandemic. These can be explained on the one hand by the increased number of visitors. Some participants complained about crowding. Additional to the increased number of visitors, the unfamiliar requirement of "keeping a distance" might also have intensified the feeling of crowding. On the other hand, these new conflicts reflect the different motives for visiting the forest described above: Above all, some of those who want to be alone and enjoy the peace and quiet feel disturbed by the number of other people in the forest.

In Italy [73], the COVID-19 pandemic has induced dramatic effects on the population of the industrialized north, whereas it has not heavily affected inhabitants of the southern regions. This might be explained in part by human exposure to high levels of fine particulate matter in the air of northern Italy, thus exacerbating the mortality. Since trees mitigate air pollution by intercepting matters onto plant surfaces and bolster the human immune system by emitting bioactive volatile organic compounds, we hypothesize a protective role of evergreen forested areas in southern Italy. In silico docking studies of potentially protective compounds found in Laurus nobilis L., a typical Mediterranean plant, were performed to search for potential antivirals. Roviello et al. [73] found that the pandemic's severity was generally lower in southern regions, especially those with more than 0.3 hectares of forest per capita. The lowest mortality rates were found in southern Italy, mainly in regions like Molise (0.007%) and Basilicata (0.005%) where the forest per capita ratio is higher than 0.5 Ha/person. Our findings suggest that evergreen Mediterranean forests and shrubland plants could have protected the southern population by emission of immuno-modulating volatile organic compounds and provision of dietary sources of bioactive compounds. Overall, results highlight the importance of nature conservation and applications to the search for natural antivirals.

The practice of Shinrin-Yoku or Forest Bathing is an outdoor therapeutic modality with mounting evidence suggesting positive effects on individuals' psychological wellbeing and overall health. Changes in state anxiety, negative affect, positive affect and state mindfulness were assessed. Results show significant increases in positive affect, vigour, friendship, and mindfulness, and decreases in negative affect, anxiety, anger, fatigue, tension, and depressive mood. Effect sizes observed for all the outcomes were significant and large, ranging from $d = 1.02$ to $d = 2.61$ [74].

The government policy to prevent the transmission of COVID-19, which affects both the social and economic aspects of society, is inevitable [75]. The available forest resources are the main alternatives for the communities around the forest that have no option to maintain their families' survival. The utilization of the timber forest and land use will impact the forest's sustainability, directly or indirectly. The prevention efforts undertaken by limiting access to forest areas are not the best alternatives during the COVID-19 pandemic.

The SARS-CoV-2-related pandemic has been the cause of some of the direst challenges that mankind has had to face in the modern era. A much more sedentary lifestyle, an

increase in unhealthy eating habits as well as mental health degradation have all combined to result in an impaired immune function and an enhanced risk of infection. To reverse these inadvertent consequences, moderate but sustained exercise is a feasible way of improving both physical as well as mental health in a time when anxiety and social isolation is on the rise. Important factor to keep in mind when looking at the benefits of a physical training program is its location. Forest bathing, or the practice of spending time outside amongst particular trees, has scientifically proven benefits when it comes to human health and the strengthening of the immune system in particular. Crucially, forests also mitigate air pollution while forming important buffer zones which lower the risks of contact with wild animals and the pathogens they harbour, such as coronaviruses. In conclusion, sport performed in green spaces bolsters the body's ability to fight off COVID-19 and other infectious diseases through the combined benefits of physical exercise and the immunostimulatory effect. With more and more evidence of the protective and purifying qualities of trees and forested areas, it is critical that efforts be made to protect current forests and increase green spaces if we are to succeed in ending the current pandemic and prevent more such devastating illnesses from making an appearance in the future [76].

Key observations [26] show that under the COVID-19 lockdown in Germany the levels of forest visits have experienced an unprecedented boom, revealing forests as a critical infrastructure for society at large. There are a number of possible reasons for this boom: people have more time available, more flexibility, more pressure at home, but also fewer alternative pastimes. People's visit patterns have shifted to the afternoons and the clear difference between the number of visitors on weekdays and weekends has substantially decreased. This strong increase and the novel user groups are strongly related to the specific COVID-19 lockdown situation. What is uncertain is if these changing use patterns will have impacts also after the lockdown, and how this could potentially impact the job of the foresters, as well as the cohabitation of different forest user groups. Given these uncertainties, future qualitative, as well as quantitative, research efforts, such as those used in this study, will be crucial for drawing conclusions about visitor preferences and motivations to visit the forest after the lockdown. The results from a nationwide survey of residents of Slovakia [77] showed that forest accessibility was a paramount factor affecting the number of forest visits in both pre-COVID-19 situations and during the pandemic. In terms of the effects on NFV, settlement size was linked with forest accessibility through distance to the nearest forest, which was lowest in villages and towns with up to several thousand inhabitants. COVID-19 and its accompanying measures had an equalizing effect on average income-number of forest visits relationships by alternatively diminishing and increasing NFV in the lowest and highest income categories, respectively. The observed pattern developed after widespread introduction of working-from-home schemes during the COVID-19 pandemic, accompanied by more flexible working schedules. Type of employment and age were revealed as additional crucial factors determining number of forest visits rates. Additionally, there were nominal increases in stated number of forest visits by respondents on parental leave and students on the one hand, while on the other hand number of forest visits rates decreased in the retired people category, probably linked with risk avoidance behavioral patterns.

Given the said levels of COVID-19 and mental health issues related [13–15,17,18], forest ecosystems services through scenery viewing, walking, riding, or jogging through the recreational areas [3,4,26,27,60–72] offer relief. Geng et al. [2] demonstrated a significant increase in the forest visit during the COVID-19. It justifies that forest services and benefits play a role in stress and mental healing.

The COVID-19 pandemic has completely changed many people's ways of living regarding freedom of movement and work. Various studies have illustrated the immense stress associated with the pandemic, regardless of age or geographical region, due to lockdown and change in routine [13–15]. The current literature showed that more research had been done on COVID-19 stress, but less on the role of forest ecosystems on COVID-19

stress and mental health. Visiting forest recreation during COVID-19 pandemics was not associated with COVID-19 cases increase in spread.

Our investigation on visitors' movement at Training Forest Enterprise Masaryk Forest Křtiny allowed us to observe that more people enjoyed the forest recreation services in March, April, and May 2021 than in previous years, with minor variations on some days. We recommend follow-up research to use visitor's recorders to thoroughly investigate the trend of forest visitors and big data on opinions on the role of forest ecosystem services, especially in the urban and suburban areas. Moreover, this data could help in deciding on future research and policy decisions regarding epidemiological situations.

Based on data analysis in previous stages of the work, the following conclusions can be formulated:

- It was found that the **peak visits are at the turn of April and May (after winter)**-regardless of external factors. In other words, the city forest is always needed for relaxation, and its proximity to the city makes it an attractive place, especially for short-term rest.
- Regardless **of the COVID-19 pandemic**, the peak of visits to the city forest falls in the following years at the turn of April and May (after winter) and remains within similar quantitative limits. In other words, the city forest is now always needed for the recreation of city people-people want to be among the greenery regardless of the pandemic.

The above basic statements allow to formulate some **general conclusions**:

- Urban society always needs contact with nature after winter in the first spring days. It is at the turn of April and May that the forest has the highest share of users. Hence, the organization of tourism and mass events is advisable in the spring. It is the best time to organize outdoor events: excursions, walks, picnics, festivals, etc. Thus, it is valuable information for the managers of the area and organizers of tourism and collective events.
- The most substantial need to improve health in Central Europe occurs in spring, after winter, when society is generally weakened and needs to be strengthened through contact with nature. The obtained results show that the human population intuitively seeks and finds ways to improve its health through recreation in the urban forest.

**Author Contributions:** Conceptualization, D.B.; methodology, D.B., J.Ł., B.F.-A.; software, J.F.; validation, D.B., J.F. and P.K.; formal analysis, D.B., J.Ł., B.F.-A.; investigation, D.B.; resources, P.K.; data preparation, D.B., J.Ł., B.F.-A.; writing—original draft preparation, D.B.; writing—review and editing, D.B., J.F., J.Ł.; visualization, J.F.; supervision, P.K.; project administration, P.K.; funding acquisition, P.K. All authors have read and agreed to the published version of the manuscript.

**Funding:** This research received no external funding.

**Institutional Review Board Statement:** Not applicable.

**Informed Consent Statement:** Not applicable.

**Data Availability Statement:** The Data attained from this study has been arhieved at Center for Open Science (OSF) Frankfurt, Germany. The work is freely open to the public for access and reuse. A Digital Object Identifier (DOI). The reference has been created as "Bamwesigye, D. Data Results Before and During COVID-19 Lockdown Crisis. https://doi.org/10.17605/OSF.IO/8NAKW (accessed on 6 December 2021).

**Acknowledgments:** The authors would like to thank dr hab. med. Jacek Łukaszkiewicz for his advisable remarks in the field of metabolism of the "sun vitamin"-vitamin $D_3$, showing its immune values to the COVID-19 virus and its therapeutic importance to the social health connected to the exposition to the UVB sun rays during the recreation in forests. Also, this article was written with the support of Lubos Kala, Director of Partnership, o.p.s. and coordinator of monitoring cyclists and pedestrians in the Czech Republic Partnership o.p.s.

**Conflicts of Interest:** The authors declare no conflict of interest.

## Appendix A

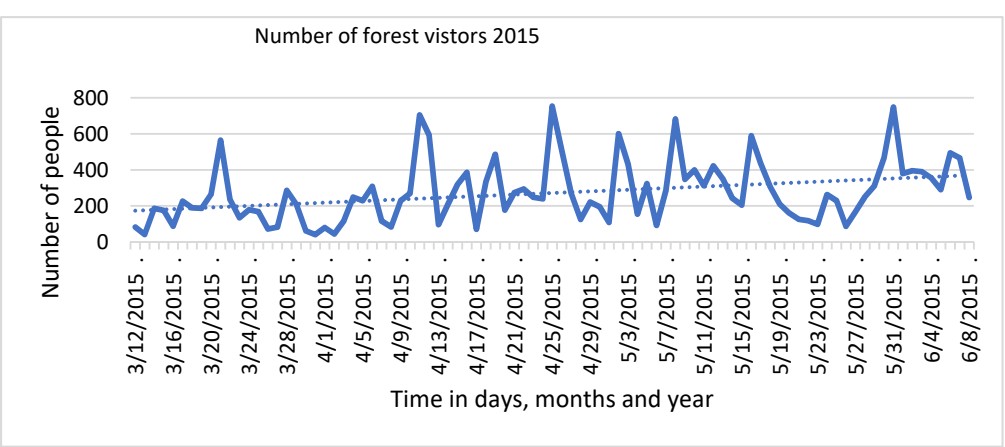

**Figure A1.** Recorded number of visitors per day in three months in 2015.

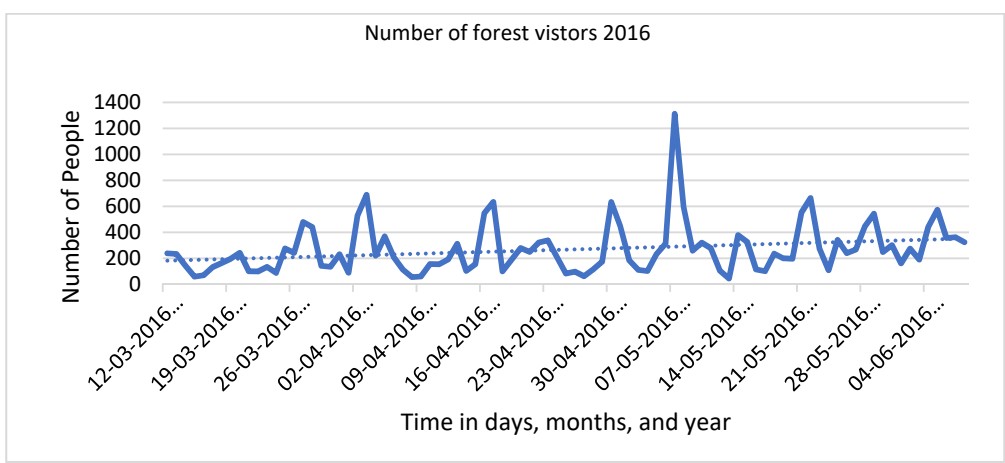

**Figure A2.** Recorded number of visitors per day in three months in 2016.

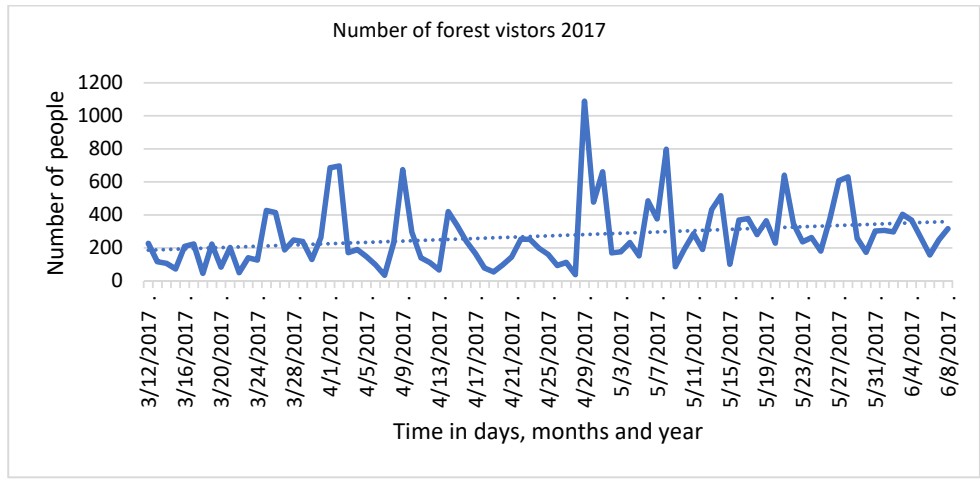

**Figure A3.** Recorded number of visitors per day in three months in 2017.

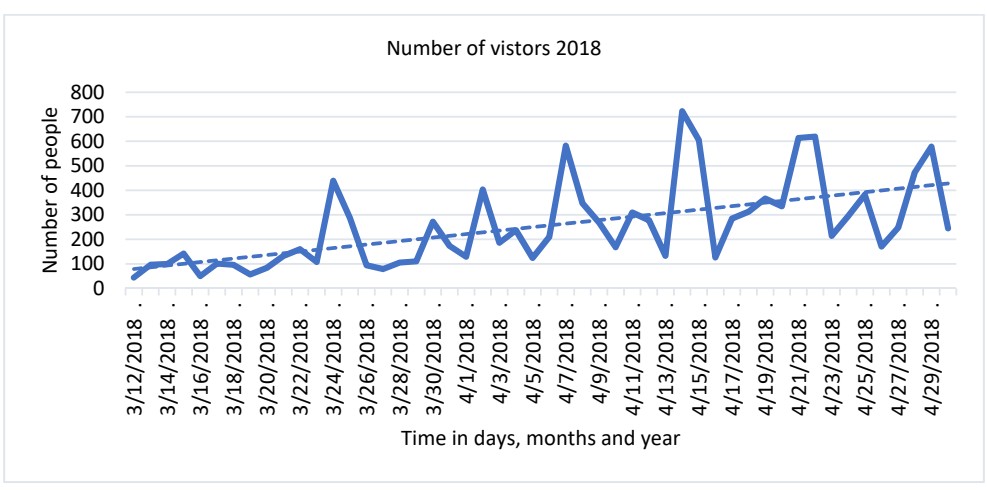

**Figure A4.** Recorded number of visitors per day in three months in 2018.

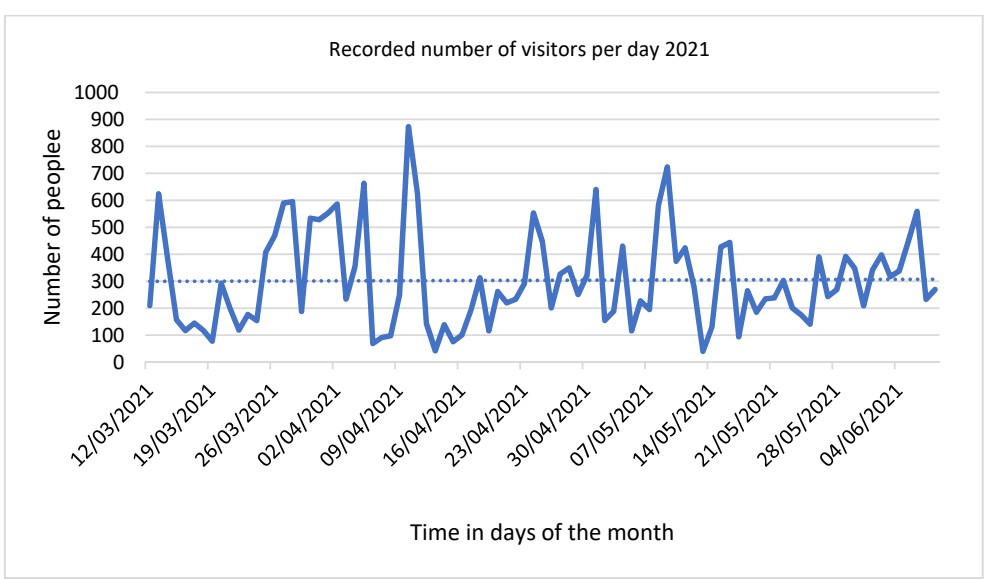

**Figure A5.** Recorded number of visitors per day in three months in 2021.

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
