# Peer review of "Forest Recreational Services in the Face of COVID-19 Pandemic Stress"

_land, doi:10.3390/land10121347_

Round 1

Reviewer 1 Report

The topic of the presented research is actual and potential for further assessment. The Paper has logical structure and clear analyses of the published sources. The results are adequate and supported by the measurements.

I would recomment to check the words divisions at the end of the rows: 18,23, 27, 55, 61, 67, 166, 219, 236, 245, 250, 338, 375, 396, 451.

Author Response

The topic of the presented research is actual and potential for further assessment. The Paper has logical structure and clear analyses of the published sources. The results are adequate and supported by the measurements.

Comment: We would like to thank a lot the reviewer for the comment.

I would recomment to check the words divisions at the end of the rows: 18,23, 27, 55, 61, 67, 166, 219, 236, 245, 250, 338, 375, 396, 451.

Comment: We have used the template from MDPI and it divides words at the end of the rows automatically.

Reviewer 2 Report

The information in the manuscript ‘Forest Recreational Services in the Face of COVID-19 Pandemic Stress’ submitted for review is new and original. In conclusion, it should be emphasized that the results of the study are presented convincingly, the proposed manuscript has a scientific value and reveals an important question of present interest – COVID-19 pandemic and recreational and culture ecosystem services provided by green areas in urban ecosystems. I recommend to the Editorial Board to accept the manuscript in present form.

Author Response

The information in the manuscript ‘Forest Recreational Services in the Face of COVID-19 Pandemic Stress’ submitted for review is new and original. In conclusion, it should be emphasized that the results of the study are presented convincingly, the proposed manuscript has a scientific value and reveals an important question of present interest – COVID-19 pandemic and recreational and culture ecosystem services provided by green areas in urban ecosystems. I recommend to the Editorial Board to accept the manuscript in present form.

Comment: We would like to thank a lot the reviewer for the comment.

Reviewer 3 Report

Article addresses an important aspect of maintaining the mental health of public representatives in an era of pandemic restrictions

Comments

The authors in the methodological part unnecessarily presented a simplified description of the infrastructure of park areas without presenting the actual area distribution including the area of the studied forest area.

The influence of density in urban areas on the number of people using green areas was not presented. No complete verification of forested area with indication of the proportion of forest stands, road lengths, and transportation factors that allow people to use forested areas.

TRAFx Infrared Trail Counter - has been implemented since March 2021 . The results of previous observations abolish a different methodology - Pyro Box Compact from Eco-counter devices were used (Data collected is from July 2014 to June 11, 2018). The traffic volume measurement system was not explained.

The blocking during the pandemic period took place from November 2020 to the end of May 2021 so the study period with the measuring device relates to a very narrow time frame of March, June.

The pandemic period was associated with a partial restriction of movement which was reflected in the change of intensification was confirmed in the study.

Chapter 5 does not define the direct effect of visiting the forest on impr

Author Response

Comments

The authors in the methodological part unnecessarily presented a simplified description of the infrastructure of park areas without presenting the actual area distribution including the area of the studied forest area.

Comment: We presented the Location, its characteristics, and data connected with the Training Forest Enterprise Masaryk Forest Krtiny – so the area is forested almost 100 %. This is where the data for study was collected.

The influence of density in urban areas on the number of people using green areas was not presented. No complete verification of forested area with indication of the proportion of forest stands, road lengths, and transportation factors that allow people to use forested areas.

 Comment: This info is not significant for this area – forest that are directly on the border of the City of Brno. We still call it urban forest that continuously is transformed in the commonly used forest. Info about the proportion of forest stands or road lenghts is not necessary and it could be extra. However the description of the study location and general conditions like you commented was freally at length.

TRAFx Infrared Trail Counter - has been implemented since March 2021 . The results of previous observations abolish a different methodology - Pyro Box Compact from Eco-counter devices were used (Data collected is from July 2014 to June 11, 2018). The traffic volume measurement system was not explained.

Comment: Counter was installed on the forest road where the barrier is installed. Limited number of cars had been passing the counters – just hunters and foresters. We added the comment in the text – see lines 266 to 283.

The blocking during the pandemic period took place from November 2020 to the end of May 2021 so the study period with the measuring device relates to a very narrow time frame of March, June. The pandemic period was associated with a partial restriction of movement which was reflected in the change of intensification was confirmed in the study.

Comment: According to some bureaucracy (connected with the COVID-19 as well) we were not able to install the counter earlier

Chapter 5 does not define the direct effect of visiting the forest on impr

Comment: The idea is probably not finished.

Reviewer 4 Report

land-1491598-peer-review-v1

Review of  Forest Recreational Services in the Face of COVID-19 Pandemic Stress

The literature review on the benefits of peri-urban green spaces and forests needs to be substantially restructured Section 3 should be. Some detail, e.g. Vitamin D etc is far too detailed and has little to do with the specifics of the study at hand. This needs to be condensed dramatically. At the same time, there is quite a bit of literature on COVID and the use of greenspaces during the pandemic but that has not been explored in enough depth.

Lines 258–278 should be expressed more succinctly.

Figure 4. It would be better to show this a single linear graph, March to July rather than as  monthly curves overlain.

Figure 5 is impossible to read/use. Split into single years and spread out vertically above each other, with months separated by thin vertical lines. The bottom axis needs fixing

I would still plot the 2018 data for what you have, and keep a gap..

Consider not plotting absolute numbers for Fig 5, but in % of each year’s March to July period. Then the relative seasonal use is better visible, and the impact pf COVID can be better shown (I would guess)

Line 335  I really like your use of ‘ceteris paribus’ but I would venture the guess that most readers would not have a clue….use the English equivalent

Figure 6, what does that tell us? What causes the low 2016 values? What other socio-economic factors, for example, come into play here? This needs to be better discussed.

Figure 7. What is A-E ?

The way the data are presented, I cannot see any impact of COVID on forest use. Thisneeds to be better presented and better discussed

Are there other mobility data that cab be drawn on for comparison?

Minor issues

The paper needs a thorough edit by a native English speaker as is numerous grammatical infelicities, awkward expressions and spelling mistakes.

Author Response

The literature review on the benefits of peri-urban green spaces and forests needs to be substantially restructured Section 3 should be. Some detail, e.g. Vitamin D etc is far too detailed and has little to do with the specifics of the study at hand. This needs to be condensed dramatically. At the same time, there is quite a bit of literature on COVID and the use of greenspaces during the pandemic but that has not been explored in enough depth.

Comment: We do not agree with shortening of Section 3. The most important and useful literature connected with our topic we added in the part of literature review.

Lines 258–278 should be expressed more succinctly.

Comment: We expressed it more succinctly – corrected – see lines 260-275.

Figure 4. It would be better to show this a single linear graph, March to July rather than as  monthly curves overlain.

Comment: March to June sigle graphs have been added in the appendix section and also described in the text. See appendix A.

Figure 5 is impossible to read/use. Split into single years and spread out vertically above each other, with months separated by thin vertical lines. The bottom axis needs fixing

Comment:  X axis adjusted and corrected acordingly.

The figure 5, yes has alot of information and as you may have read it is the most important figure in this study which can all togther show the trend analysis comparision throught the years and in given months. Sperating resulsts could show clearer figures but gives room for ambiguity for readers who are interested in forest visitor analsys and especially before and during COVID-19.

I would still plot the 2018 data for what you have, and keep a gap..

Comment: We kept 2018  in the trend analysis because of the major months of analysis (March, April,and May).

Consider not plotting absolute numbers for Fig 5, but in % of each year’s March to July period. Then the relative seasonal use is better visible, and the impact pf COVID can be better shown (I would guess)

Comment: The absolute figures too  illustrate the trend as we wanted it to. We highly appreciate your suggestion.

Line 335  I really like your use of ‘ceteris paribus’ but I would venture the guess that most readers would not have a clue….use the English equivalent

Comment: Yes and No, but we would like to keep it because it makes sense.

Figure 6, what does that tell us? What causes the low 2016 values? What other socio-economic factors, for example, come into play here? This needs to be better discussed.

Comment: True! The diffrence therein is quite marginal! And could be related to the socioeconomic factors. However, ceteris paribus, our study focused on the trend and how it relates with COVID-19. We have added this idea to our discussion.

Figure 7. What is A-E ?

Comment: (where A is 2015, B is 2016, C is 2017, D is 2018 and E is 2021), added.

The way the data are presented, I cannot see any impact of COVID on forest use. Thisneeds to be better presented and better discussed

Comment: We hypothesed that the urban forests are of extremely vital meaning for urban environments but also very more crucial  for the urban society, the qual-ity of peoples’ life and health - especially during the pandemic. For this reason, the im-portance of shaping greenery such - as urban forests are - to obtain its maximum pro-health impact is arising all over the world.

Therefore, the impact of COVID-19 on forest use as is equally vital will be considered given that it is a whole broad topic.

Are there other mobility data that cab be drawn on for comparison?

Comment: No

Minor issues

The paper needs a thorough edit by a native English speaker as is numerous grammatical infelicities, awkward expressions and spelling mistakes.

Comment: Corrected by the native speaker

Round 2

Reviewer 4 Report

The authors are to be commended on attempting the review

In my initial review I wrote:

The literature review on the benefits of peri-urban green spaces and forests needs to be substantially restructured Section 3 should be. Some detail, e.g. Vitamin D etc is far too detailed and has little to do with the specifics of the study at hand. This needs to be condensed dramatically. At the same time, there is quite a bit of literature on COVID and the use of greenspaces during the pandemic but that has not been explored in enough depth.

The authors commented:: We do not agree with shortening of Section 3. The most important and useful literature connected with our topic we added in the part of literature review.

No. This paper purports to be about " Forest Recreational Services in the Face of COVID-19 Pandemic Stress ". So, while a discussion of general benefits needs to be presented, it has to be  balanced and not throw in everything the authors know about a side issue. The Vitamin D discussion should be cut to one paragraph or 3-4 sentences.

At the same time paper still does not engage with the literature on green space use during the COVID-19 pandemic. This is only touched on, but should be a corner stone of the paper. It is is not the role of the reviewer to spell  out this literature in detail. As as a result of the lack of action in this section, I am forced to change my recommendations and request a MAJOR revision before the paper can be considered for publication

==================

My comments to figure 4 still stand. Contrary to the author's assertions, they have not been shown individually in the appendix. The appendix shows figure 5 as individual curves,

Figure 5 is still an unreadable mess. If the  authors want to persist with it, then at the very least make the plotted lines thinner (1pt) so that the are more discrete and readable. The same applies to all graphs

==================

The authors response in the revised text to my comments on the 2016 data is non sensical

==================

The discussion section is not not very balanced. The impact of OVID had not been adequately discussed. It shows the inability or unwillingness of the authors to engage with the extensive literature on green space use during the COVID-19 pandemic. 

==================

The inclusion of the individual graphs is welcomed and improves the understanding, but their presentation is really unprofessional

  1. These graphs need to be standardised in height, ie same y axis, 0-1400
    The x axis labels needs to have same orientation throughout, 
  2. To improve readability, remove year from x axis labels (as that is included in the caption)
  3.  remove the stray dots from x axis labels
  4. Make sure the labels are the same dat pattern (some are 3/28 others are 26-03 for ex).
  5. Make sure the tick mark increments are the same for all graphs

Make the graph lines thinner. they are too thick

==================

Line 481  achieved  should be archived

==================

The paper still needs full professional edit by a NATIVE English speaker. It is still full of grammatical mistakes and problems with expression

Author Response

Review 4 – second round

The authors are to be commended on attempting the review

In my initial review I wrote:

The literature review on the benefits of peri-urban green spaces and forests needs to be substantially restructured Section 3 should be. Some detail, e.g. Vitamin D etc is far too detailed and has little to do with the specifics of the study at hand. This needs to be condensed dramatically. At the same time, there is quite a bit of literature on COVID and the use of greenspaces during the pandemic but that has not been explored in enough depth.

The authors commented: We do not agree with shortening of Section 3. The most important and useful literature connected with our topic we added in the part of literature review.

No. This paper purports to be about " Forest Recreational Services in the Face of COVID-19 Pandemic Stress ". So, while a discussion of general benefits needs to be presented, it has to be  balanced and not throw in everything the authors know about a side issue. The Vitamin D discussion should be cut to one paragraph or 3-4 sentences.

Comment: we have extensively shorted the section and also shifted it to make it section 2. It is still longer than you recommended but it was reduced by 75% of the literature on Vitamin D. The remaining paragraphs give a clear relationship among covid-19, forest recreation services, and vitamin D.

The correlation is that if people were accepted to be out for some time outside during covid-19 lockdown, besides relieving stress, there is also a possibility to enjoy Vitamin D, which boosts immunity during covid-19.

Section highlighted.

At the same time paper still does not engage with the literature on green space use during the COVID-19 pandemic. This is only touched on, but should be a corner stone of the paper. It is is not the role of the reviewer to spell  out this literature in detail. As as a result of the lack of action in this section, I am forced to change my recommendations and request a MAJOR revision before the paper can be considered for publication

Comment: We have corrected the mistake; in the introduction, we have made a chronological flow of the related topics i.e., covid-19 & mental health stress, urban and forest recreation.

My comments to figure 4 still stand. Contrary to the author's assertions, they have not been shown individually in the appendix. The appendix shows figure 5 as individual curves,

Comment: Corrected

Figure 5 is still an unreadable mess. If the  authors want to persist with it, then at the very least make the plotted lines thinner (1pt) so that the are more discrete and readable. The same applies to all graphs

Comment: We really tried to do our best and we hope that now will reviewer be satisfied.

The discussion section is not not very balanced. The impact of OVID had not been adequately discussed. It shows the inability or unwillingness of the authors to engage with the extensive literature on green space use during the COVID-19 pandemic. 

Comment: We really tried to do our best and we hope that now will reviewer be satisfied now.

The inclusion of the individual graphs is welcomed and improves the understanding, but their presentation is really unprofessional

  1. These graphs need to be standardised in height, ie same y axis, 0-1400
    The x axis labels needs to have same orientation throughout, 
  2. To improve readability, remove year from x axis labels (as that is included in the caption)
  3.  remove the stray dots from x axis labels
  4. Make sure the labels are the same dat pattern (some are 3/28 others are 26-03 for ex).
  5. Make sure the tick mark increments are the same for all graphs

Make the graph lines thinner. they are too thick

Comment: We really tried to do our best and we hope that now will reviewer be satisfied.

Line 481  achieved  should be archived

Comment: corrected

The paper still needs full professional edit by a NATIVE English speaker. It is still full of grammatical mistakes and problems with expression

Comment: The texts had been checked by native speaker again and we really appreciate his help. We hope that now the texts are better and more readable.